# Novel Perbutyrylated Glucose Derivatives of (–)-Epigallocatechin-3-Gallate Inhibit Cancer Cells Proliferation by Decreasing Phosphorylation of the EGFR: Synthesis, Cytotoxicity, and Molecular Docking

**DOI:** 10.3390/molecules26144361

**Published:** 2021-07-19

**Authors:** Ya Wang, Xiao-Jing Shen, Fa-Wu Su, Yin-Rong Xie, Li-Xia Wang, Ning Zhang, Yi-Long Wu, Yun Niu, Dong-Ying Zhang, Cheng-Ting Zi, Xuan-Jun Wang, Jun Sheng

**Affiliations:** 1Key Laboratory of Pu-er Tea Science, Ministry of Education, College of Science, Yunnan Agricultural University, Kunming 650201, China; wangya9188@126.com (Y.W.); xyr351527080@163.com (Y.-R.X.); wanglixia0315@163.com (L.-X.W.); zn1193594574@163.com (N.Z.); ynau_wuyl@126.com (Y.-L.W.); Niuyun0429@163.com (Y.N.); Zhangdongying365@163.com (D.-Y.Z.); 2College of Food Science and Technology, Yunnan Agricultural University, Kunming 650201, China; 3Party Committee of Organ, Yunnan Agricultural University, Kunming 650201, China; sxj243@sina.com; 4State Key Laboratory for Conservation and Utilization of Bio-Resources in Yunnan, Yunnan Agricultural University, Kunming 650201, China; su_faw@126.com

**Keywords:** EGCG, synthesis, cytotoxicity, EGFR, molecular docking

## Abstract

Lung cancer is one of the most commonly occurring cancer mortality worldwide. The epidermal growth factor receptor (EGFR) plays an important role in cellular functions and has become the new promising target. Natural products and their derivatives with various structures, unique biological activities, and specific selectivity have served as lead compounds for EGFR. D-glucose and EGCG were used as starting materials. A series of glucoside derivatives of EGCG (**7**–**12**) were synthesized and evaluated for their in vitro anticancer activity against five human cancer cell lines, including HL-60, SMMC-7721, A-549, MCF-7, and SW480. In addition, we investigated the structure-activity relationship and physicochemical property–activity relationship of EGCG derivatives. Compounds **11** and **12** showed better growth inhibition than others in four cancer cell lines (HL-60, SMMC-7721, A-549, and MCF), with IC_50_ values in the range of 22.90–37.87 μM. Compounds **11** and **12** decreased phosphorylation of EGFR and downstream signaling protein, which also have more hydrophobic interactions than EGCG by docking study. The most active compounds **11** and **12**, both having perbutyrylated glucose residue, we found that perbutyrylation of the glucose residue leads to increased cytotoxic activity and suggested that their potential as anticancer agents for further development.

## 1. Introduction

Lung cancer is one of the most commonly occurring cancer mortality worldwide, accounting for 20% of all cancer-related deaths [1]. Approximately 75–85% of lung cancers are non-small-cell lung cancer (NSCLC), including lung adenocarcinoma, squamous cell carcinoma (SCC), and large cell carcinoma (LCC) histological subtypes, and are typically diagnosed at advanced stages [2,3]. Recently, the identification of key oncogenic-driver mutations has led to the development of molecular targeted therapies for patients with NSCLC [4]. The epidermal growth factor receptor (EGFR) plays an important role in cellular functions, and approximately 50% of lung cancer patients possess EGFR overactivated [5]. Thus, EGFR tyrosine kinase has become an attractive therapeutic target for NSCLC [6,7,8].

EGFR tyrosine kinase inhibitors (TKI), such as gefitinib and erlotinib are used in clinical therapy to treat NSCLC [9,10,11,12]. However, the T790M mutation of EGFR has led to resistance to these EGFR TKIs [13], and new drugs have been developed, such as afatinib [10,14] and osimertinib [15]. Natural products and their derivatives with various structures, unique biological activities, and specific selectivity have served as lead compounds for EGFR. For example, aeroplysinin selectively inhibits EGFR [16]; shikonin preferentially inhibits cell proliferation by inhibiting EGFR-signaling [17].

Green tea (*Gamellia sinensis* leaves) is an extremely popular drink in the world, and many studies show the positive effects of green tea on cancer, including lung cancer [18,19]. It is generally agreed that tea catechins were first isolated from green tea, making it a promising chemopreventive agent [20]. The major catechins in green tea leaves are (-)-epicatechin (EC, **1**), (-)-epicatechin-3-gallate (ECG, **2**), (-)-epigallocatechin (EGC, **3**), and (-)-epigallocatechin-3-gallate (EGCG, **4**) (Figure 1). Among these catechins, EGCG is the most studied due to its abundance, constituting 50–80% of total catechins content in green tea [21], and has been reported to have stronger physiological activities than others [22,23,24]. Several studies have indicated that treatment with EGCG inhibits tumor incidence such as skin, lung, liver, breast, prostate, and stomach [25]. There is considerable evidence that EGCG inhibits tumorigenesis, signal transduction pathway, cell invasion, angiogenesis, and metastasis [26,27,28,29,30]. Notably, previous studies reported that EGCG inhibits the activation of EGFR and human epidermal growth factor receptor 2 (HER2) in human lung cancer cells [31,32], and EGCG inhibits the tyrosine kinase activity of EGFR in human A431 epidermoid carcinoma cells [33].

However, the use of EGCG often has limitations such as easy oxidation [34], low bioavailability [35], and easy hydrolyses by bacterial and possibly host esterases [36]; thus, attempts to use EGCG in the treatment of human neoplasia have been mostly unsuccessful. To obtain more potent analogs and overcome these problems, many EGCG derivatives have been synthesized, including methyl-protected EGCG [37], acetyl-protected EGCG, [38], and EGCG monoester derivatives [39], and most of them exhibited more potent activity than EGCG. Recently, the glycoconjugates of small molecule anticancer drugs have become an attractive strategy in order to improve drug efficacy and pharmacokinetics, in addition to reducing side effects [40,41,42]. In our previous study, we thus reported that the synthesized glucosylated EGCG derivatives exhibited enhanced cytotoxicity and were more stable [5,43].

Since butyrate is a well-known histone deacetylase (HDAC) inhibitor, and its anticancer effect shows promising therapeutic potential [44], acetyl-protected EGCG analogs exhibited more activity than EGCG [37]. In this study, we synthesized a series of glucoside derivatives of EGCG (**7**–**12**) and evaluated for their in vitro anticancer activity against five human cancer cell lines, including HL-60 (leukemia), SMMC-7721 (hepatoma), A-549 (lung cancer), MCF-7 (breast cancer), and SW480 (colon cancer). In addition, Western blotting and molecular docking analyses of these compounds were studied and correlated with their anticancer activity.

## 2. Results and Discussion

### 2.1. Chemistry

The synthesis of the (-)-epigallocatechin-3-gallate glucoside derivatives **7**–**12** was performed according to the reaction pathways illustrated in Scheme 1 and Scheme 2. D-glucose and EGCG were used as starting materials. 2,3,4,6-Tetra-O-butyryl-α-D-glucopyranosyl bromide **6** was prepared using the method reported in the literature [45]. EGCG (**4**) was allowed to react with the above compound **6** in the presence of K_2_CO_3_ in acetone at 55 °C for 12 h to obtain peracetyl EGCG glucoside derivatives **7** and **8** in 25–30% yields (Scheme 1). Then, compounds **7** and **8** were treated with KOH solution in CH_3_OH at 0 °C for 72 h to yield EGCG glucoside derivatives **9** and **10** in 52–55% yields [27]. Finally, the free glucose of compounds **9** and **10** was buturylated with butyric anhydride in pyridine solution at 0 °C for 12 h to yield perbutyryl EGCG glucoside derivatives **11** and **12** in 80–82% yields (Scheme 2).

All the synthesized compounds were characterized by ^1^H-NMR, ^13^C-NMR, electrospray ionization mass spectrometry (ESI-MS), and high-resolution mass spectrometry (HRESI-MS). In the ^1^H-NMR spectra, the formation of the D-glucose residue was confirmed by the resonance of C^1^‴-H/C^1^⁗-H signal (*δ* 4.54–4.97 ppm) with the coupling constant of the anomeric proton of D-glucose residue (*J*_1_‴/*J*_2_⁗ = 7.6 Hz–7.8 Hz) (Appendix A). In our early study, the coupling constant of the anomeric proton of the glucose residue (*J*_1−2_) is typically <4.0 Hz for the α–linkage and >7.6 Hz for the β–linkage [46].

The chemical structures of compounds were further confirmed by 2D-NMR spectra data. Based on the HMBC data (Figure 2), the strong coupling occurred between C^1^⁗-H (*δ* 4.97 ppm) and C-4″ (137.3 ppm) of the D ring from the HMBC of compound **7**. Similarly, coupling occurred between C^1^⁗-H (*δ* 4.54 ppm) and C-4″ (*δ* 138.5 ppm) of the D ring and also between C^1^‴-H (*δ* 4.67 ppm) of the other glucosyl residue and C-4′ (*δ* 137.9 ppm) of the B ring from the HMBC of compound **8**. The chemical shift for the proton at C^4^-H exhibited coupling with C-2, C-3; C^2^′-H, C^6^″-H occurred with C-2 and C^2^″-H, C^6^″-H occurred with C-11; and C^2^-H occurred with C-11 in both compounds **7** and **8**. These results indicated that the β-peracetylated glucoside (H-1⁗-H→C-4″) had attached to the EGCG scaffold in compound **7**, and two β-peracetylated glucosides (H-1‴-H→C-4′ and H-1⁗-H→C-4″) had attached in compound **8**.

### 2.2. Evaluation of Anticancer Activity

All EGCG derivatives **7**–**12** were evaluated for their cytotoxicity against five human cancer cell lines, including HL-60 (leukemia), SMMC-7721 (hepatoma), A-549 (lung cancer), MCF-7 (breast cancer), and SW480 (colon cancer). EGCG (**4**) and cisplatin were taken as positive control compounds. The screening procedure was based on the standard 3-(4,5-dimethyl-thiazol-2-yl)-2,5-diphenyltetrazolium bromide (MTT) method [47]. Their activities were expressed by the IC_50_ value (concentration of drug inhibiting 50% cell growth), and the data are presented in Table 1.

As shown in Table 1, the majority of glucoside derivatives of EGCG display weak cytotoxicity against all five cancer cell lines, and most of them show higher potency than the control EGCG. Among the glucoside derivatives of EGCG, compounds **11** and **12** showed better growth inhibition than others in four cancer cell lines (HL-60, SMMC-7721, A-549, and MCF), with IC_50_ values in the range of 22.90–37.87 μM. Interestingly, all glucoside derivatives of EGCG showed good cytotoxicity against A-549 cells except compound **10** (having IC_50_ > 40 μM). Previously, we reported that the cytotoxic activity of glucosylated EGCG derivatives with a free glucose residue mostly shows weak activity against breast cancer (MCF and MDA-MB-231 cells) [27]. However, our findings indicate that EGCG derivatives with a perbutyrylated glucose residue (**11** and **12**) lead to increased activity rather than a free glucose residue (**11** vs. **9** and **12** vs. **10**).

Cancer chemotherapy is often associated with low/non-selectivity of cancer drugs, which attack cancer cells and normal cells, leading to serious side effects. EGCG derivatives **11** and **12** were tested for their growth inhibitory effects on normal human bronchial epithelial cells (BEAS-2B) (Table 1). The IC_50_ values of compounds **11** and **12** (all having IC_50_ > 40 μM), indicated that glucoside derivatives of EGCG showed moderated selectivity toward cancer cells.

### 2.3. EGCG Derivatives Inhibited Phosphorylation of EGFR and Downstream Signaling Protein in A-549 Cells

To investigate whether the effects of EGCG derivatives **11** and **12** might involve EGFR signaling, we assessed the expression of several key regulators that function within the EGFR signaling pathway. The results showed that treatment with EGCG slightly exhibited P-EGFR, P-AKT, and P-Erk (Figure 3A). Compounds **11** and **12** further exhibited the phosphorylation of EGFR and downstream signaling protein, while the total protein level of EGFR remained unchanged (Figure 3B,C). As shown in Figure 3D,E, it can be observed that EGCG, **11** and **12** can reduce the expression of phosphorylated EGFR, ERK, and AKT proteins. In summary, EGFR signaling may play a significant role in the effects of EGCG derivatives **11** and **12** in A-549 cells.

### 2.4. Molecular Docking Analysis

To further investigate the potential binding between EGFR and the compounds, molecular docking was performed. The binding modes of compounds **4**, **11,** and **12** at the ATP-binding pocket of EGFR (PDB code: 2ITY) were shown in Figure 4. As shown in Figure 4, compounds **11** and **12** could be docked into the active site of EGFR, the butyryl group occupies the ATP-binding pocket. The detailed docking results were presented in Appendix A. It was observed that the binding energies between EGFR and glucoside derivatives of EGCG **11** and **12** are weak, compared with EGCG. The docking energy for compounds **4**, **11,** and **12** were −5.18 kcal/mol, 0.47 kcal/mol, and 1.81 kcal/mol, respectively. The complex structures for compounds **11** and **12** were presented in Figure 5. The key residues were labeled and the important molecular interaction including the hydrogen bonds (black dotted lines), hydrophobic interactions (red dotted lines), π-stacking (gray dotted lines), and salt bridges (yellow dotted lines) were summarized in Table 2.

As shown in Figure 5, the molecules in complex with EGFR were analyzed in Figure 5A,B. It was observed that there are three hydrophobic interactions between compound **11** and three residues of EGFR (PHE723, ALA743, LEU844), five hydrogen bonds between **11** and three residues of EGFR (LEU718, MET793, ASN842), and two salt bridges between **11** and two residues of EGFR (LYS745, ARG841). There are seven hydrophobic interactions between compound **12** and seven residues of EGFR (ALA722, PHE723, VAL726, LYS745, MET793, ARG841, and VAL876), three hydrogen bonds between **12** and three residues of EGFR (GLU758, ASP855, and VAL876), one π-stacking and a salt bridge between **12** and the residue of EGFR (LYS745 and ARG841, respectively). Similarly, EGCG was docked into the EGFR are shown in Appendix A, which result in ten hydrogen bonds, one hydrophobic interaction, one π-stacking and one salt bridge, respectively. The presence of more hydrophobic interactions in compounds **11** and **12** seems the key factor for their high activity.

## 3. Experimental

### 3.1. Materials and Methods

D-glucose and *n*-butyric anhydride were purchased from Aladdin Chemical Co., Ltd. (Guangzhou, China); (-)-epigallocatechin-3-gallate was obtained from Chengdu Proifa Technology Development Co., Ltd. (Chengdu, China); boron trifluoride etherate was obtained from J&K Chemical Technology Co., Ltd. (Beijing China); 3-(4,5-dimethyl-thiazol-2-yl)-2,5-diphenyltetrazolium bromide (MTT) was purchased from Sigma-Aldrich (St. Louis, MO, USA). All reagents were commercially available and used without further purification unless indicated otherwise. The melting points were measured by an X-4 melting point apparatus and were uncorrected. Optical rotations were obtained with a Jasco P-1020 Automatic Digital Polariscope MS data were obtained in the ESI mode on API Qstar Pulsar instrument; HRMS data were obtained in the ESI mode on LCMS-IT-TOF (Shimadzu, Kyoto, Japan); ^1^H-NMR and ^13^C-NMR spectra were recorded on Bruker DRX-500 (Bruker BioSpin GmbH, Rheinstetten, Germany) instruments, using tetramethylsilane (TMS) as an internal standard: chemical shifts (*δ*) are given in ppm, coupling constants (*J*) in Hz, the solvent signals were used as references (CD_3_OD: *δ*_C_ = 49.0 ppm; residual CH_3_OH in CD_3_OD: *δ*_H_ = 4.78 ppm). Column chromatography (CC): silica gel (200–300 mesh; Qingdao Makall Group CO., LTD; Qingdao; China). All reaction was monitored using thin-layer chromatography (TLC) on silica gel plates, which was visualized by ultraviolet light (254 nm) and/or 10% phosphomolybdic acid/EtOH. The human cancer cell lines (HL-60, SMMC-7721, A-549, MCF-7, and SW480) were obtained from a Shanghai cell bank in China.

### 3.2. Chemistry

General procedure for the synthesis of 2,3,4,6-tetra-*O*-acetyl-β-d-glucopyranosyl (-)-epigallocatechin-3-gallates (**7** and **8**)

A solution of EGCG (1.8 g, 4 mmol) was dissolved in acetone (5 mL), and potassium carbonate (K_2_CO_3_, 1.1 g, 8 mmol) was added with stirring 30 min, and 2,3,4,6-tetra-*O*-acetyl-α-d-glucopyranosyl bromide **6** (1.6 g, 4 mmol) was added with heating and stirring at 55 °C for 12 h. The mixture was cooled and filtered, the filtrate was concentrated under vacuum, and the residue was purified by column chromatography in silica gel (CHCl_3_/CH_3_OH, 15:1 → 9:1) to afford the major product.

[4″-O-(2⁗,3⁗,4⁗,6⁗-tetra-*O*-acetyl-β-d-glucopyranosyl)]-(-)-epigallocatechin-3-gallate (**7**).

White powder; yield 25%; mp. 100 °C to 101 °C; ^1^H-NMR (CD_3_OD, 500 MHz) *δ* 6.91 (s, 2H, C^2^″, C^6^″-H), 6.49 (s, 2H, C^2^′, C^6^′-H), 5.95 (s, 2H, C^6^-H, C^8^-H), 5.54 (brs, 1H, C^2^-H), 5.34 (t, 1H, *J* = 9.0 Hz), 5.24–5.22 (m, 1H, C^3^-H), 5.21 (m, 1H), 5.09 (t, 1H, *J* = 9.0 Hz), 4.97 (d, 1H, *J* = 9.0 Hz, C^1^⁗-H), 4.28 (dd, 1H, *J* = 5.4 Hz, 12.4 Hz), 4.04 (dd, 1H, *J* = 2.4 Hz, 12.4 Hz), 3.88–3.84 (m, 1H), 2.98 (dd, 1H, *J* = 4.6 Hz, 17.6 Hz, C^4^-H_a_), 2.86–2.83 (m, 1H, C^4^-H_b_), 2.04–1.99 (m, 12H, 4× OCH_3_); ^13^C-NMR (CD_3_OD, 125 MHz) *δ* 172.5, 171.6, 171.6, 171.3, 166.8 (C=O), 157.9 (C-5), 157.8 (C-9), 157.2 (C-7), 151.7 (C-3′, C-5′), 146.7 (C-3″, C-5″), 137.3 (C-4″), 133.7 (C-4′), 130.7 (C-1′), 128.2 (C-1″), 110.1 (C-2″, C-6″), 106.7 (C-2′, C-6′), 102.7 (C-1⁗), 99.2 (C-10), 96.5 (C-6), 95.8 (C-8), 78.4 (C-2), 74.1, 73.2, 72.9, 70.4 (C-3), 69.7, 62.9 (C-6⁗), 26.8 (C-4), 20.8 (OCH_3_), 20.6 (OCH_3_), 20.5 (OCH_3_), 20.5 (OCH_3_); ESIMS: *m/z* 811 [M + Na]^+^.

[4′-O-(2‴,3‴,4‴,6‴-tetra-*O*-acetyl-β-d-glucopyranosyl)-4″-*O*-(2⁗,3⁗,4⁗,6⁗-tetra-*O*-acetyl-β-d-glucopyranosyl)]-(-)-epigallocatechin-3-gallate (**8**)

White powder; yield 30%; mp. 108 °C to 110 °C; ^1^H-NMR (CD_3_OD, 600 MHz) *δ* 6.92 (s, 2H, C^2^″, C^6^″-H), 6.58 (s, 2H, C^2^′, C^6^′-H), 5.98 (s, 1H, C^3^-H), 5.96 (s, 2H, C^6^-H, C^8^-H), 5.55 (brs, 1H, C^2^-H), 5.04 (s, 2H), 4.67 (d, 1H, *J* = 9.5 Hz, C^1^⁗-H), 4.54 (d, 1H, *J* = 9.0 Hz, C^1^‴-H), 4.41–4.36 (m, 2H), 4.28–4.24 (m, 2H), 3.59–3.55 (m, 2H), 3.51–3.40 (m, 2H), 3.36–3.31 (m, 2H), 3.00 (dd, 1H, *J* = 2.4 Hz, 12.6 Hz, C^4^-H_a_), 2.89–2.86 (m, 1H, C^4^-H_b_), 2.08–2.00 (m, 24H, 8× OCH_3_); ^13^C-NMR (CD_3_OD, 150 MHz) *δ* 173.1, 173.1, 173.1, 173.1, 173.0, 173.0, 173.0, 173.0, 166.9 (C=O), 158.0 (C-5), 157.9 (C-9), 157.0 (C-7), 151.6 (C-3′, C-5′), 151.3 (C-3″, C-5″), 138.5 (C-4″), 137.9 (C-4′), 134.2 (C-1′), 128.5 (C-1″), 110.4 (C-2″, C-6″), 107.3 (C-1⁗), 107.1 (C-2′, C-6′), 107.0 (C-1‴), 99.3 (C-6, C-8), 96.8 (C-10), 78.2 (C-2), 77.6, 77.5, 76.2, 76.1, 75.1, 75.0, 71.6, 71.5, 70.7 (C-3), 64.6 (C-6‴), 64.5 (C-6⁗), 26.8 (C-4), 20.9 (OCH_3_), 20.9 (OCH_3_), 20.9 (OCH_3_), 20.9 (OCH_3_), 20.8 (OCH_3_), 20.8 (OCH_3_), 20.8 (OCH_3_), 20.8 (OCH_3_); ESIMS: *m*/*z* 1141 [M + Na]^+^.

General procedure for the synthesis of β-d-glucopyranosyl (-)-epigallocatechin-3-gallates (**9** and **10**)

To a solution of 2,3,4,6-tetra-*O*-acetyl-β-d-glucopyranosyl (-)-epigallocatechin-3-gallates **7**/**8** (0.1 mmol) in CH_3_OH (2 mL), a potassium hydroxide solution (0.1 mmol, dissolved in CH_3_OH) was added. The mixture was stirred at 0 °C for 72 h and then neutralized with Dowex 50WX4-400 ion-exchange resin to pH = 7. The solvent was evaporated under vacuum, and the residue was purified by column chromatography in silica gel (CHCl_3_/CH_3_OH, 4:1) to afford the product.

[4″-*O*-(β-d-glucopyranosyl)]-(-)-epigallocatechin-3-gallate (**9**)

White powder; yield 55%; mp. 118 °C to 120 °C; ^1^H-NMR (CD_3_OD, 400 MHz) *δ* 6.90 (s, 2H, C^2^″, C^6^″-H), 6.59 (s, 2H, C^2^′, C^6^′-H), 6.02 (d, 1H, *J* = 2.2 Hz, C^6^-H), 6.00 (d, 1H, *J* = 2.2 Hz, C^8^-H), 5.73 (brs, 1H, C^3^-H), 5.12 (s, 1H, C^2^-H), 4.69 (d, 1H, *J* = 9.0 Hz, C^1^⁗-H), 3.81–3.78 (m, 1H), 3.74–3.72 (m, 1H), 3.49–3.41 (m, 4H), 3.03–3.01 (m, 1H, C^4^-H_a_), 2.95–2.92 (m, 1H, C^4^-H_b_); ^13^C-NMR (CD_3_OD, 100 MHz) *δ* 166.4 (C=O), 158.2 (C-7), 157.9 (C-5), 157.8 (C-9), 151.6 (C-3′, C-5′), 146.7 (C-3″, C-5″), 138.6 (C-4″), 133.7 (C-4′), 130.7 (C-1′), 127.6 (C-1″), 110.2 (C-2″, C-6″), 108.2 (C-1⁗), 106.9 (C-2′, C-6′), 98.3 (C-10), 97.1 (C-6), 95.7 (C-8), 78.4, 77.6, 76.3, 75.1, 70.6, 68.6 (C-2), 61.8 (C-6⁗), 26.5 (C-4); ESIMS: *m*/*z* 619 [M−H]^–^.

[4′-*O*-(β-d-glucopyranosyl)-4″-*O*-(β-d-glucopyranosyl)]-(-)-epigallocatechin-3-gallate (**10**)

White powder; yield 52%; mp. 125 °C to 126 °C; ^1^H-NMR (CD_3_OD, 400 MHz) *δ* 6.92 (s, 2H, C^2^″, C^6^″-H), 6.57 (s, 2H, C^2^′, C^6^′-H), 5.98 (d, 1H, *J* = 2.3 Hz, C^6^-H), 5.96 (d, 1H, *J* = 2.3 Hz, C^8^-H), 5.55 (brs, 1H, C^3^-H), 5.11 (s, 1H, C^2^-H), 4.67 (d, 1H, *J* = 9.5 Hz, C^1^⁗-H), 4.54 (d, 1H, *J* = 9.5 Hz, C^1^‴-H), 3.83 (m, 6H), 3.48–3.41 (m, 6H), 2.99–2.89 (dd, 1H, *J* = 2.4 Hz, 12.6 Hz, C^4^-H_a_), 2.89–2.86 (m, 1H, C^4^-H_b_); ^13^C-NMR (CD_3_OD, 100 MHz) *δ* 166.8 (C=O), 157.9 (C-5), 157.8 (C-9), 156.9 (C-7), 151.5 (C-3′, C-5′), 151.2 (C-3″, C-5″), 138.4 (C-4″), 137.6 (C-1′), 134.1 (C-4′), 128.3 (C-1″), 110.3 (C-2″, C-6″), 107.8 (C-1⁗), 107.0 (C-1‴), 106.9 (C-2′, C-6′), 99.1 (C-10), 96.6 (C-695.8 (C-8), 78.5, 78.4, 78.4, 78.3, 78.1 (C-3), 77.5, 77.5, 75.1, 75.1, 70.6, 70.5, 70.4 (C-2), 61.8 (C-6⁗), 61.8 (C-6‴), 26.7 (C-4); ESIMS: *m*/*z* 781 [M−H]^–^.

General procedure for the synthesis of 2,3,4,6-tetra-*O*-butyryl-β-d-glucopyranosyl (-)-epigallocatechin-3-gallates (**11** and **12**)

To a solution of β-d-glucopyranosyl (-)-epigallocatechin-3-gallates **9**/**10** (0.05 mmol) in pyridine (1 mL), butyric anhydride (0.1 mL, 0.5 mmol) was added, and the stirring was continued at 0 °C for 12 h. The reaction mixture was diluted with CH_2_Cl_2_ (5 mL) and washed with aqueous saturated NaHCO_3_ solution, the organic layer was then dried with Na_2_SO_4,_ and the solvent was evaporated under vacuum; the residue was purified by column chromatography in silica gel (CHCl_3_/CH_3_OH, 15:1) to afford the product.

[4″-*O*-(2⁗,3⁗,4⁗,6⁗-tetra-*O*-butyryl-β-d-glucopyranosyl)]-(-)-epigallocatechin-3-gallate (**11**)

White powder; yield 82%; mp. 92 °C to 94 °C; ^1^H-NMR (CD_3_OD _3_, 400 MHz) *δ* 6.91 (s, 2H, C^2^″, C^6^″-H), 6.48 (s, 2H, C^2^′, C^6^′-H), 5.94 (s, 2H, C^6^-H, C^8^-H), 5.54 (brs, 1H, C^2^-H), 5.31 (m, 1H), 5.24–5.22 (m, 1H, C^3^-H), 5.21 (m, 1H), 5.10 (t, 1H, *J* = 9.0 Hz), 4.97 (d, 1H , *J* = 9.0 Hz, C^1^⁗-H), 4.29 (dd, 1H, *J* = 5.4 Hz, 12.4 Hz), 4.06–4.03 (m, 1H), 3.88–3.84 (m, 1H), 2.98 (dd, 1H, *J* = 4.6 Hz, 17.6 Hz, C^4^-H_a_), 2.86–2.83 (m, 1H, C^4^-H_b_), 2.21–2.18 (m, 8H, 4× COCH_2_), 1.63–1.55 (m, 8H, 4 × CH_2_CH_3_), 0.94–0.88 (m, 12 H, 4 × CH_2_CH_3_); ^13^C-NMR (CD_3_OD, 100 MHz) *δ* 172.4, 171.6, 171.6, 171.2, 166.8 (C=O), 157.9 (C-5), 157.8 (C-9), 157.1 (C-7), 151.6 (C-3′, C-5′), 146.7 (C-3″, C-5″), 137.2 (C-4″), 133.7 (C-4′), 130.7 (C-1′), 128.1 (C-1″), 110.0 (C-2″, C-6″), 106.6 (C-2′, C-6′), 102.7 (C-1⁗), 99.2 (C-10), 96.5 (C-6), 95.8 (C-8), 78.2 (C-2), 74.0, 73.1, 72.9, 70.4 (C-2), 69.7, 62.8 (C-6⁗), 35.9 (*C*OCH_2_), 35.8 (*C*OCH_2_), 35.7 (*C*OCH_2_), 35.6 (*C*OCH_2_), 26.7 (C-4), 18.2 (*C*H_2_CH_3_), 18.2 (*C*H_2_CH_3_), 18.1 (*C*H_2_CH_3_), 18.1 (*C*H_2_CH_3_), 13.6 (CH_2_*C*H_3_), 13.6 (CH_2_*C*H_3_), 13.5 (CH_2_*C*H_3_), 13.5 (CH_2_*C*H_3_); ESIMS: *m*/*z* 923 [M + Na]^+^.

[4′-*O*-(2‴,3‴,4‴,6‴-tetra-*O*-butyryl-β-d-glucopyranosyl)-4″--(2⁗,3⁗,4⁗,6⁗-tetra-*O*-butyryl-β-d-glucopyranosyl)]-(-)-epigallocatechin-3-gallate (**12**)

White powder; yield: 80%; mp. 95 °C to 97 °C; ^1^H-NMR (CD_3_OD, 600 MHz) *δ* 6.92 (s, 2H, C^2^″, C^6^″-H), 6.58 (s, 2H, C^2^′, C^6^′-H), 5.98 (s, 1H, C^3^-H), 5.96 (s, 2H, C^6^-H, C^8^-H), 5.55 (brs, 1H, C^2^-H), 5.03 (s, 2H), 4.68 (d, 1H, *J* = 9.5 Hz, C^1^⁗-H), 4.56 (d, 1H, *J* = 9.5 Hz, C^1^‴-H), 4.42–4.38 (m, 2H), 4.28–4.26 (m, 2H), 3.60–3.56 (m, 2H), 3.50–3.40 (m, 2H), 3.36–3.32 (m, 2H), 3.01 (dd, 1H, *J* = 2.4 Hz, 12.6 Hz, C^4^-H_a_), 2.88–2.86 (m, 1H, C^4^-H_b_), 2.60–2.58 (m, 4H, 2× COCH_2_), 2.35–2.20 (m, 12H, 6 × COCH_2_), 1.62–1.60 (m, 4H, 2× CH_2_CH_3_), 1.58–1.56 (m, 12H, 6× CH_2_CH_3_), 1.04–0.93 (m, 6H, 2× CH_2_CH_3_), 0.91–0.93 (m, 18H, 6× CH_2_CH_3_); ^13^C-NMR (CD_3_OD, 150 MHz) *δ* 173.1, 173.1, 173.1, 173.1, 173.0, 173.0, 173.0, 173.0, 166.9 (C=O), 158.0 (C-5), 157.9 (C-9), 157.0 (C-7), 151.6 (C-3′, C-5′), 151.3 (C-3″, C-5″), 138.4 (C-4″), 137.9 (C-4′), 134.1 (C-1′), 128.5 (C-1″), 110.3 (C-2″, C-6″), 107.8 (C-1⁗), 107.1 (C-1‴), 107.0 (C-2′, C-6′), 99.2 (C-6), 99.2 (C-8), 95.9 (C-10), 78.2 (C-2), 77.5, 77.5, 76.1, 76.1, 75.0, 74.9, 71.5, 71.5, 70.6 (C-3), 64.5 (C-6⁗), 64.5 (C-6‴), 35.9 (*C*OCH_2_), 35.8 (*C*OCH_2_), 35.8 (*C*OCH_2_), 35.7 (*C*OCH_2_), 35.7 (*C*OCH_2_), 35.7 (*C*OCH_2_), 35.6 (*C*OCH_2_), 35.6 (*C*OCH_2_), 26.8 (C-4), 18.2 (*C*H_2_CH_3_), 18.2 (*C*H_2_CH_3_), 18.2 (*C*H_2_CH_3_), 18.2 (*C*H_2_CH_3_), 18.2 (*C*H_2_CH_3_), 18.1 (*C*H_2_CH_3_), 18.1 (*C*H_2_CH_3_), 18.1 (*C*H_2_CH_3_), 13.6 (CH_2_*C*H_3_), 13.6 (CH_2_*C*H_3_), 13.5 (CH_2_*C*H_3_), 13.5 (CH_2_*C*H_3_), 13.5 (CH_2_*C*H_3_), 13.5 (CH_2_*C*H_3_), 13.4 (CH_2_*C*H_3_), 13.4 (CH_2_*C*H_3_); ESIMS: *m*/*z* 1365 [M + Na]^+^.

### 3.3. Cytotoxicity Assay

The effects of the (-)-epigallocatechin-3-gallate glucoside derivatives on the survival of cancer cells were determined using the MTT assay. Five human cancer cell lines (HL-60, SMMC-7721, A-549, MCF-7, and SW480) were used in the cytotoxicity assay. All the cancer cells were seeded in 96-well plates and then each tumor cell line was exposed to the test compound at various concentrations in triplicate for 48 h. After the incubation, MTT (100 μg) was added to each well, and the incubation continued for 4 h at 37 °C. After removal of the culture medium, the produced MTT formazan crystals were dissolved with 150 μL DMSO and measured at 492 nm using a microplate reader. The percentage of inhibition was calculated as follows: inhibition ratio (IR, %) = (1 − OD(sample)/OD(control)) × 100%. The experiments were carried out in triplicate, and the IC_50_ (the concentration of drug that inhibits cell growth by 50%) values were determined.

### 3.4. Western Blotting

A-549 cells were lysed in RIPA buffer containing PMSF (protease and phosphatase inhibitors) and quantified via BCA protein assay. Proteins separated on 8% SDS–PAGE electrophoresis and then blotted onto polyvinyl difluoride membranes. After the membranes were blocked with BSA for 1 h, the expression of various proteins was detected using primary and secondary antibodies conjugated with horseradish peroxidase. HRP was detected using the Prolight HRP Chemiluminescent Kit (Tiangen Biotech, Beijing, China) and FluorChem E Synthem (ProteinSimple, Santa Clara, CA, USA).

### 3.5. Docking Studies

The X-ray crystal structure of EGCG (PDB code: 2ITY) was retrieved from Protein Data Bank (www.rcsb.org (accessed on 17 February 2020)). Ligand docking was carried out by applying the Lamarckian Genetic Algorithm implemented in AutoDock 4.2. The modeled EGCG structure was employed as the receptor to docking with these small molecules. DiscoveryStudio 4.0 software was used for the preparation of ligand and receptor. Autodock Tools 1.5.6 was used for grid and docking according to the literature [48] The grid size was set to 60 Å, 60 Å, and 60 Å along the X-, Y- and Z-axis to recognize the binding site. The default 0.375 Å spacing was adopted for the grid box. The number of GA runs was set to 50, and the maximum number of evals (medium) was set to 5,000,000 on AutoGrid 4.2.6 and AutoDock 4.2.6. In the selected cluster, conformations with the lowest binding energy and RMSD (<2.0 Å) were finally chosen to analyze the receptor–ligand interaction. Other miscellaneous parameters were assigned to the default values obtained from the AutoDock 4.2.6.

## 4. Conclusions

In conclusion, we synthesized a series of glucoside derivatives of EGCG (**7**–**12**) and evaluated for their in vitro anticancer activity against five human cancer cell lines, including HL-60, SMMC-7721, A-549, MCF-7, and SW480. All these derivatives display different levels of anticancer activity, which can be affected by the nature of substituents on the glucose residue. Among them, compounds **11** and **12** showed better growth inhibition than others in four cancer cell lines (HL-60, SMMC-7721, A-549, and MCF), with IC_50_ values in the range of 22.90–37.87 μM. EGCG derivatives **11** and **12** also display moderate selectivity toward cancer cells over normal human bronchial epithelial cells (BEAS-2B). The docking study indicated that the presence of more hydrophobic interactions in compounds **11** and **12**. In addition, compounds **11** and **12** decreased phosphorylation of EGFR and downstream signaling protein. The most active compounds **11** and **12**, both having perbutyrylated glucose residue, led us to conclude that perbutyrylation of the glucose residue causes increased cytotoxic activity, which is indicative of their potential as anticancer agents for further development.

## Data Availability

The data presented in this study are available on request from the corresponding author.

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
