# Peer review of "Novel Perbutyrylated Glucose Derivatives of (–)-Epigallocatechin-3-Gallate Inhibit Cancer Cells Proliferation by Decreasing Phosphorylation of the EGFR: Synthesis, Cytotoxicity, and Molecular Docking"

_molecules, 2021, doi:10.3390/molecules26144361_

Round 1

Reviewer 1 Report

The manuscript by Wang et al, entitled “Novel per-butyrylated glucoses derivatives of (-)-epigallocatechin-3-gallate inhibit cancer cells proliferation by decreasing phosphorylation of the EGFR: Synthesis, cytotoxicity and molecular docking” where they evaluated anticancer effect of (-)-epigallocatechin-3-gallate derivative against several cancer cell lines. The work is superficially done, the claims are not supported by the data.

  1. Most of the compounds are not active their Ic50 values are close to 40uM.
  2. Partial inhibition on p-EGFR/AKT/ERK. The total of phosphorylated is missing. Quantitation of blots would give better picture.
  3. It would be nice to compare epigallocatechin-3-gallate along with derivatives.

Author Response

Dear Reviewer,

 Thank you for your comments about our manuscript entitled “Novel per-butyrylated glucoses derivatives of (-)-epigallocatechin-3-gallate inhibit cancer cells proliferation by decreasing phosphorylation of the EGFR: Synthesis, cytotoxicity and molecular docking ”. We believe that our manuscript has now improved by addressing to the comments from you. We have amended the manuscript accordingly, please see the following point-to-point response.

Comments:

The manuscript by Wang et al, entitled “Novel per-butyrylated glucoses derivatives of (-)-epigallocatechin-3-gallate inhibit cancer cells proliferation by decreasing phosphorylation of the EGFR: Synthesis, cytotoxicity and molecular docking” where they evaluated anticancer effect of (-)-epigallocatechin-3-gallate derivative against several cancer cell lines. The work is superficially done, the claims are not supported by the data.

Most of the compounds are not active their Ic50 values are close to 40 uM.

---Thank you very much for your comments and suggestions. To identify novel per-butyrylated glucoses derivatives of (-)-epigallocatechin-3-gallate, we treated five human cancer cell lines (HL-60, SMMC-7721, A-549, MCF-7 and SW480) with EGCG and its derivatives (40 μM) and measured the cell viability using the MTT assay. The glucoside derivatives of EGCG display weak cytotoxicity against all five cancer cells. Among them, compounds 11 and 12 showed better growth inhibition than others.

Partial inhibition on p-EGFR/AKT/ERK. The total of phosphorylated is missing. Quantitation of blots would give better picture.

---Thank you very much for your comments and pointing it out. We have made correction according to the reviewer’s comments. For the western blot results, we revised and added new electrophoresis results based on the reviewers’ comments, including the total amount of p-EGFR/AKT/ERK phosphorylation and quantitative figure in Figure 3.

It would be nice to compare epigallocatechin-3-gallate along with derivatives.

---Thank you very much for your comments and suggestions. Epigallocatechin-3-gallate (EGCG) was taken as control compounds in the part of cytotoxicity assay, western blotting, and docking studies.

Reviewer 2 Report

The manuscript describing “Novel per-butyrylated glucoses derivatives of (-)-epigallocatechin-3-gallate inhibit cancer cells proliferation by decreasing phosphorylation of the EGFR: Synthesis, cytotoxicity and molecular docking” by Ya Wang et al. is a combined experimental and computational (docking) study on a series of epigallocatechin-3-gallate derivatives. The study is based on a strategy stating that glycoconjugates of small-molecule anticancer drugs can be potentially more bioavailable and more resistant to the enzymatic inactivation by either cancer cells or the host organism. I am an expert not on the biochemical, but rather on the computational part of the study, however I think that the investigations are valuable and competently carried out. The manuscript is however not presented in an easy-to-read way, and I recommend a revision that is minor with regard to the scientific value, but significant with respect to the presentation and text.

A thorough review of the language and style is necessary (probably also with the help of the MDPI-associated staff), because in many places the small grammar and vocabulary mistakes add up to hamper the reading of the interesting manuscript. I will give only a few short examples:

line 72: Figure 1 has wrong caption (from the MDPI template);

line 73: “is often limitations” should be “often has limitations”;

line 74: “attempt” should be “attempts”;

line 77: “have synthesized” should be “have been synthesized”;

line 82: it should read “EGCG derivatives exhibited enhanced cytotoxicity and were more stable”;

line 98: “has been” is not necessary;

line 133: “five human cancer cells” should probably be “five human cancer cell lines” or similar;

line 148: “we have been reported” should be “we have reported”;

lines 195-196: “the chemical structures of compounds 11 and 12 were performed” – I cannot follow this sentence;

line 214: another section on “Results and discussion”? Please correct;

line 369: not AutoDock Tools 1.56 but 1.5.6,

and so on. Please carry out a thorough spell- and style-check of the manuscript.

The methodological issues are rather small and hopefully easy to correct and explain:

line 135: the compound 4, EGCG, is – according to Table 1 – not strongly active against the studied cancer cell lines. Why then is it called a “positive control”?

Molecular docking analysis, section 2.4: please observe that the predicted “best docking energies” for 11 and 12 are positive, suggesting lack of docking capacity. Also the predicted inhibition constants are much worse for 11 and 12 than for 4 (EGCG), which strangely was not found strongly active towards the cell lines. Does it mean that, after all, the EGFR cannot be strongly implicated in the mechanism? Please comment.

Beginning of line 334: is it really (C=0) with “zero”, or should it be “O” for oxygen?

Line 370: No units are given for the grid size (60 – should be Angstroms).

End of reviewer remarks

Author Response

Dear Reviewer,

 Thank you for your comments about our manuscript entitled “Novel per-butyrylated glucoses derivatives of (-)-epigallocatechin-3-gallate inhibit cancer cells proliferation by decreasing phosphorylation of the EGFR: Synthesis, cytotoxicity and molecular docking ”. We believe that our manuscript has now improved by addressing to the comments from you. We have amended the manuscript accordingly, please see the following point-to-point response.

Comments:

The manuscript describing “Novel per-butyrylated glucoses derivatives of (-)-epigallocatechin-3-gallate inhibit cancer cells proliferation by decreasing phosphorylation of the EGFR: Synthesis, cytotoxicity and molecular docking” by Ya Wang et al. is a combined experimental and computational (docking) study on a series of epigallocatechin-3-gallate derivatives. The study is based on a strategy stating that glycoconjugates of small-molecule anticancer drugs can be potentially more bioavailable and more resistant to the enzymatic inactivation by either cancer cells or the host organism. I am an expert not on the biochemical, but rather on the computational part of the study, however I think that the investigations are valuable and competently carried out. The manuscript is however not presented in an easy-to-read way, and I recommend a revision that is minor with regard to the scientific value, but significant with respect to the presentation and text.

A thorough review of the language and style is necessary (probably also with the help of the MDPI-associated staff), because in many places the small grammar and vocabulary mistakes add up to hamper the reading of the interesting manuscript. I will give only a few short examples:

line 72: Figure 1 has wrong caption (from the MDPI template);

line 73: “is often limitations” should be “often has limitations”;

line 74: “attempt” should be “attempts”;

line 77: “have synthesized” should be “have been synthesized”;

line 82: it should read “EGCG derivatives exhibited enhanced cytotoxicity and were more stable”;

line 98: “has been” is not necessary;

line 133: “five human cancer cells” should probably be “five human cancer cell lines” or similar;

line 148: “we have been reported” should be “we have reported”;

lines 195-196: “the chemical structures of compounds 11 and 12 were performed” – I cannot follow this sentence;

line 214: another section on “Results and discussion”? Please correct;

line 369: not AutoDock Tools 1.56 but 1.5.6,

and so on. Please carry out a thorough spell- and style-check of the manuscript.

---Thank you very much for your comments and suggestions. The manuscript was read by professor Zi-Hua Jiang worked for many years in Lakehead University in Canada, and we already changed the inappropriate description.

The methodological issues are rather small and hopefully easy to correct and explain:

line 135: the compound 4, EGCG, is – according to Table 1 – not strongly active against the studied cancer cell lines. Why then is it called a “positive control”?

---Thank you very much for pointing it out. We already changed the inappropriate description.

Molecular docking analysis, section 2.4: please observe that the predicted “best docking energies” for 11 and 12 are positive, suggesting lack of docking capacity. Also the predicted inhibition constants are much worse for 11 and 12 than for 4 (EGCG), which strangely was not found strongly active towards the cell lines. Does it mean that, after all, the EGFR cannot be strongly implicated in the mechanism? Please comment.

---Thank you very much for pointing out it. EGCG is one of the most abundant and biologically active compound in green tea. A previous study reported that EGCG inhibits the activation of EGFR [1], and Liang et al. reported that EGCG binds to and inhibits the tyrosine kinase activity of EGFR in human A431 epidermoid carcinoma cells [2]. In this research, a series of glucoside derivatives of EGCG (712) and evaluated for their in vitro anticancer activity against five human cancer cell lines, including HL-60 (leukemia), SMMC-7721 (hepatoma), A-549 (lung cancer), MCF-7 (breast cancer) and SW480 (colon cancer). The larger spatial structures of compounds 11 and 12 including four butyryl parts and eight butyryl parts, respectively, lead to lack of docking capacity. However, butyrate in the compounds 11 and 12 is a histone deacetylase (HDAC) inhibitor resulted in per-butyrylated glucoses derivatives of EGCG inhibit cancer cells proliferation.

[1] Ma YC, Li C, Gao F, Xu Y, Jiang ZB, Liu JX, Jin LY. Epigallocatechin gallate inhibits the growth of human lung cancer by directly targeting the EGFR signaling pathway. Oncol. Rep. 2014, 31 (3), 1343–1349.

[2] Liang YC, Lin-Shiau SY, Chen CF, Lin JK. Suppression of extracellular signals and cell proliferation through EGF receptor binding by (-)-epigallocatechin gallate in human A431 epidermoid carcinoma cells. J. Cell Biochem. 1997, 67 (1), 55.

Beginning of line 334: is it really (C=0) with “zero”, or should it be “O” for oxygen?

Line 370: No units are given for the grid size (60 – should be Angstroms).

---Thank you very much for your comments. We already changed the inappropriate description.

Reviewer 3 Report

In the manuscript titled “Novel per-butyrylated glucoses derivatives of (-)-epigallocatechin-3-gallate inhibit cancer cells proliferation by decreasing phosphorylation of the EGFR: Synthesis, cytotoxicity and molecular docking”, Wang et al. synthesized several EGCG derivatives and performed in vitro anticancer activity assays against five human cancer cell lines. In addition, they used molecular modeling to further characterize the binding of some selected compounds.

  1. In the evaluation of the anticancer activity of the derived compounds, the authors reported the values of the IC50s that are in the 22-40 mM range.  These values seem relatively high to be considered as anticancer therapeutics as the authors claimed the majority of the derivatives display weak cytotoxicity against the cancer cells even if they show a higher potency compared to the control EGCG. Also, their best compounds show a weak to moderate selectivity towards the cancel cells. How do the authors reconcile this?
  2. What do the authors mean by “The results showed that treatment with EGCG slightly exhibited P-EGFR,P-AKT, and P-Erk”?
  3. The authors evaluated the binding energies of compounds 11 and 12 and found them to be positive. The control had a negative binding energy. In fact, the best docking energy given in Table S2 for the compound 11 seems incorrect (It must be around -8.1+(-0.4)-(-10.9) = +1.5 kcal mol) This implies that the compounds are quietly weakly interacting with EGFR. How do the authors reconcile this? With the positive interaction energy values, further analysis seems redundant.
  4. What are the error bars of IC50s?
  5. The word “depravities” in several places must be “ derivatives”.

Author Response

Dear Reviewer,

 Thank you for your comments about our manuscript entitled “Novel per-butyrylated glucoses derivatives of (-)-epigallocatechin-3-gallate inhibit cancer cells proliferation by decreasing phosphorylation of the EGFR: Synthesis, cytotoxicity and molecular docking ”. We believe that our manuscript has now improved by addressing to the comments from you. We have amended the manuscript accordingly, please see the following point-to-point response.

Comments:

In the manuscript titled “Novel per-butyrylated glucoses derivatives of (-)-epigallocatechin-3-gallate inhibit cancer cells proliferation by decreasing phosphorylation of the EGFR: Synthesis, cytotoxicity and molecular docking”, Wang et al. synthesized several EGCG derivatives and performed in vitro anticancer activity assays against five human cancer cell lines. In addition, they used molecular modeling to further characterize the binding of some selected compounds.

  1. In the evaluation of the anticancer activity of the derived compounds, the authors reported the values of the IC50s that are in the 22-40 mM range. These values seem relatively high to be considered as anticancer therapeutics as the authors claimed the majority of the derivatives display weak cytotoxicity against the cancer cells even if they show a higher potency compared to the control EGCG. Also, their best compounds show a weak to moderate selectivity towards the cancel cells. How do the authors reconcile this?

---Thank you very much for your comments. It can be seen from Table 1 that compounds 11 and 12 showed weak cytotoxicity compared with the positive control (cisplatin). However, EGCG is one of the most abundant and biologically active compound in green tea and EGCG has limitations such as easy oxidation, low bioavailability, easy hydrolyses by bacterial and possibly host esterases. For this reason, we synthesized a series of per-butyrylated glucoses derivatives of EGCG and tested their cytotoxicity against five human cancer cell lines. The results showed that compounds 11 and 12 were higher cytotoxicity than EGCG. In addition, compounds 11 and 12 showed different selectivity to cancer cells and their activities may be different.

  1. What do the authors mean by “The results showed that treatment with EGCG slightly exhibited P-EGFR,P-AKT, and P-Erk”?

---Thank you very much for your comments and suggestions. We supplemented experiments and verified that EGCG and its derivatives can reduce the expression of p-EGFR, p-Akt, and p-ERK proteins (Figure 3).

  1. The authors evaluated the binding energies of compounds 11 and 12 and found them to be positive. The control had a negative binding energy. In fact, the best docking energy given in Table S2 for the compound 11 seems incorrect (It must be around -8.1+(-0.4)-(-10.9) = +1.5 kcal mol) This implies that the compounds are quietly weakly interacting with EGFR. How do the authors reconcile this? With the positive interaction energy values, further analysis seems redundant.

---Thank you very much for your comments and suggestions. We already changed the errors the inappropriate description in Table S2. EGCG inhibits the activation of EGFR has reported in many previously studies [1,2]. The larger spatial structures of compounds 11 and 12 including four butyryl parts and eight butyryl parts, respectively, lead to lack of docking capacity. However, butyrate in the compounds 11 and 12 is a histone deacetylase (HDAC) inhibitor resulted in per-butyrylated glucoses derivatives of EGCG inhibit cancer cells proliferation.

[1] Ma YC, Li C, Gao F, Xu Y, Jiang ZB, Liu JX, Jin LY. Epigallocatechin gallate inhibits the growth of human lung cancer by directly targeting the EGFR signaling pathway. Oncol. Rep. 2014, 31 (3), 1343–1349.

[2] Liang YC, Lin-Shiau SY, Chen CF, Lin JK. Suppression of extracellular signals and cell proliferation through EGF receptor binding by (-)-epigallocatechin gallate in human A431 epidermoid carcinoma cells. J. Cell Biochem. 1997, 67 (1), 55.

  1. What are the error bars of IC50s?

---Thank you very much for pointing it out. We already added the error bars of IC50s in the Table 1.

  1. The word “depravities” in several places must be “ derivatives”.

—Thank you very much for pointing out the errors. We already changed the errors description.

Round 2

Reviewer 1 Report

The authors made the appropriate revisions.